# Discrepancies in Antimicrobial Susceptibility between the JP2 and the Non-JP2 Genotype of *Aggregatibacter actinomycetemcomitans*

**DOI:** 10.3390/antibiotics11030317

**Published:** 2022-02-27

**Authors:** Margareta Granlund, Carola Höglund Åberg, Anders Johansson, Rolf Claesson

**Affiliations:** 1Department of Clinical Microbiology, Umeå University, S-90187 Umeå, Sweden; margra136@gmail.com; 2Division of Molecular Periodontology, Department of Odontology, Umeå University, S-90187 Umeå, Sweden; carola.hoglund.aberg@umu.se (C.H.Å.); anders.johansson@umu.se (A.J.); 3Division of Oral Microbiology, Department of Odontology, Umeå University, S-90187 Umeå, Sweden

**Keywords:** *Aggregatibacter actinomycetemcomitans*, JP2 genotype, antimicrobial susceptibility, wild-type, MIC distribution

## Abstract

The *Aggregatibacter actinomycetemcomitans* JP2 genotype is associated with high leukotoxin production and severe (aggressive) periodontitis. The aim of this study was to compare the antimicrobial susceptibility of JP2 and non-JP2 genotype strains. Minimal inhibitory concentrations (MICs) of 11 antimicrobials were determined for 160 *A. actinomycetemcomitans* of serotype a, b, or c, mostly isolated in Sweden or Ghana. MIC distributions for benzylpenicillin and fusidic acid revealed a more susceptible subpopulation for 38 serotype b strains, including the 32 of the JP2 genotype, with a benzylpenicillin MIC range of 0.125–0.5 mg/L. In contrast, benzylpenicillin MIC ≤ 16 mg/L was the estimated 99.5% epidemiological cutoff (ECOFF) of all strains. Beta-lactamase production was not detected. The fusidic acid MIC distribution of 11 strains of *Aggregatibacter aphrophilus* agreed with that found in non-JP2 strains. Cefotaxime, meropenem, levofloxacin, and trimethoprim–sulfamethoxazole MICs were all ≤0.25 mg/L, while MIC_90_ values for amoxicillin, azithromycin and tetracycline were 1 mg/L. Metronidazole MICs varied between 0.5 and >256 mg/L. The discrepant findings indicate that *A. actinomycetemcomitans* may be divided into two separate wild types, with a suggested intrinsic reduced susceptibility for benzylpenicillin in the majority of non-JP2 genotype strains. Possible implications for the treatment of *A. actinomycetemcomitans* infections are discussed.

## 1. Introduction

*Aggregatibacter actinomycetemcomitans* is a member of the HACEK group, which encompasses *Haemophilus* species, *Aggregatibacter* species, *Cardiobacterium hominis, Eikenella corrodens*, and *Kingella* species [1]. These fastidious Gram-negative bacteria are normal inhabitants of the upper respiratory tract in humans but can also cause invasive infections such as infective endocarditis (IE) and abscesses [1,2,3]. In a Danish study of bacteremia, *Haemophilus* species other than *H. influenzae* were the most common bacteria found within the HACEK group, followed by *Aggregatibacter* spp. [4]. This is in accordance with that found in a recent Swedish study in which *Eikenella corrodens* and *H*. *parainfluenzae* had the highest bacteremic incidence of the members in the HACEK group. However, *A. actinomycetemcomitans* was the species most prone to cause IE, with figures higher than previously reported for viridans streptococci and *Staphylococcus aureus* [5].

More commonly described infections associated with *A. actinomycetemcomitans* are periodontal diseases [6,7], with a rapidly progressive localized form primarily seen in adolescents [8,9]. In the new classification system based on the staging and grading of periodontitis, the most severe forms of periodontitis are scored stage III–IV [10].

A specific clone of *A. actinomycetemcomitans*, the JP2 genotype, was initially detected in individuals of African descent, but is now reported worldwide [11,12]. Individuals colonized with the JP2 genotype of *A. actinomycetemcomitans* have a pronounced increased risk of developing periodontitis [13,14]

The JP2 genotype is characterized by a 530 deletion within the leukotoxin gene promoter region and produces a substantially higher amount of leukotoxin than other strains [15,16].

The leukotoxin has the capacity to kill human leukocytes and is suggested to initiate a modulation of the local immune system, resulting in an overgrowth of a range of so-called periodontitis-associated bacterial species [9]. In addition, the killing of monocytes results in the release of i.a. IL-1β, a cytokine which stimulates the activation of bone-degrading osteoclasts [17], which may lead to the loss of teeth seen even in young individuals [16]. *A. actinomycetemcomitans* with a full-length leukotoxin promoter region is described as the non-JP2 genotype. Additional differences between the two genotypes have been reported, among them variations in the genome content and the presence of distinct genomic islands [18,19]. *A. actinomycetemcomitans* can be divided into seven serotypes (a–g) and the JP2 genotype belongs to serotype b [16]. Phylogenetic analyses of the population structure of *A. actinomycetemcomitans* made by whole-genome sequencing with different selections of strains and outgroups has, by large, identified three major lineages associated with separate clades of serotypes with serotype b and c grouped in the same lineage [20,21].

Various studies have reported antibiotic susceptibility data for *A. actinomycetemcomitans* [22,23,24,25,26,27], but with the exception of the extensive study of amoxicillin susceptibility in *A. actinomycetemcomitans* by Jensen and co-workers [28], data for the JP2 genotype are scarcely reported. A comprehensive overview of the antimicrobial susceptibility of this species is difficult to achieve due to differences in antimicrobial susceptibility (AST) techniques used, and since the susceptibility is often reported in relation to divergent breakpoints, a fact that may lead to interpretation differences. Antimicrobial clinical breakpoints are determined by breakpoints committees, and constitute concentrations of antibiotics that for an identified bacterial infection will guide in choice of antibiotic treatments [29,30]. Many factors are taken into account in the process of setting breakpoints, for example, pharmacokinetic/pharmacodynamics (PK/PD) relationships for the antimicrobial agents, the type and location of the infection, dosing regiments, toxicology, resistance mechanisms and wild-type MIC distributions. The definition of both clinical breakpoints and non-species-related PK/PD breakpoints is an ongoing process in the European Committee on Antimicrobial Susceptibility Testing (EUCAST), and changes in the breakpoints are published yearly on the website [31].

To overcome the lack of specific clinical breakpoints for *Aggregatibacter*, the breakpoints for *Haemophilus influenzae* and/or for anaerobic Gram-negative bacteria have commonly been used. Since one of the many aspects to consider when breakpoints are defined is the documentation of clinical outcome for the specific bacteria and antimicrobial agent in question [29,30], the use of breakpoints designed for other species can be deceptive if used in a clinical context to guide therapeutic options. Non-species-related (PK/PD) breakpoints presented for a selection of antimicrobial agents at the EUCAST website [31] can to some extent facilitate the evaluation of antibiotic susceptibility data [32]. Another aid to the evaluation of AST results, recommended in EUCAST general documents on organisms and agents without breakpoints, is MIC distributions and agent/species-specific epidemiological cut-off values (ECOFF) presented on the website [31]. The ECOFFs are breakpoints defined to distinguish microorganisms with acquired resistance mechanisms from the so-called wild-type organisms, but these breakpoints do not imply any statement regarding clinical susceptibility or the likelihood of therapeutic success. The defined wild-type bacteria can be intrinsically resistant to a specific agent, i.e., inborn resistant without an acquired resistant mechanism [30,31]. No MIC distributions or ECOFF values are yet presented for *A. actinomycetemcomitans*; EUCAST 2022, v 12.0. In this study, we examined the susceptibility of JP2 and non-JP2 genotype *A. actinomycetemcomitans* for 11 antimicrobial agents and analyzed MIC values in relation to MIC ranges and MIC distributions.

## 2. Results

Antimicrobials for parenteral treatment and potential agents for the peroral treatment of *A. actinomycetemcomitans* were examined. For cefotaxime, meropenem, levofloxacin, and trimethoprim–sulfamethoxazole, all the studied *A. actinomycetemcomitans* strains had low MICs, at or below 0.25 mg/L (Table 1). The MIC_90_-values for amoxicillin, azithromycin and tetracycline were 1 mg/L, and for gentamicin 2 mg/L, indicating that 90% of the strains were inhibited by these concentrations. The MIC range for gentamicin was 0.25–4 mg/L with an MIC_50_ of 1 mg/L (Table 1), which renders less than 50% of the tested strains susceptible according to the PK/PD breakpoints of 0.5/0.5 mg/L. Since the MIC values of the population were tightly grouped this breakpoint would thus divide a probable wild-type distribution of *A. actinomycetemcomitans*. It is notable that the clinical gentamicin breakpoints for *Enterobacterales* (according to EUCAST breakpoint tables2022, v 12.0) [31] are 2/2 mg/L, which points to the considerations made when the clinical breakpoints are defined.

The PK/PD breakpoint for benzylpenicillin susceptibility is ≤0.25 mg/L, which would render 94% (30 of 32) of the tested JP2 genotype strains susceptible, but only 4.7% (6 of 128) of the non-JP2 strains (Table 2, Figure 1a). No wild-type distributions are presented for *Aggregatibacter* at the EUCAST website. Four antimicrobials: metronidazole, benzylpenicillin, fusidic acid, and azithromycin, showed wide MIC ranges and/or high MIC values (Table 1). In Figure 1, we present MIC distributions for these antimicrobials (Figure 1). Wide MIC ranges usually indicate heterogenous populations that most commonly are caused by the presence of a subpopulation with an acquired resistance mechanism in addition to a susceptible or intrinsically resistant wild-type population [30,31]. As noted in the MIC distributions, two subpopulations were present for benzylpenicillin and fusidic acid (Figure 1a,b). However, here a possible wild-type distribution was separated from a more susceptible population mostly consisting of the JP2 genotype strains. Interestingly, a small population of six serotype b non-JP2 strains also had low MICs for benzylpenicillin and fusidic acid (Figure 1a,b). The ECOFFinder analysis showed an estimated ECOFF at 95% and 99.5%, of 8 mg/L and 16 mg/L, respectively, for benzylpenicillin against *A. actinomycetemcomitans* (Figure 2). This defined a possible wild-type distribution with MIC ≤ 16 mg/L. In concordance with what is seen in Figure 1a, a population with a low MIC distribution is separated from the vast majority of the strains. Beta-lactamase production was not found in any of the *A. actinomycetemcomitans* strains of the study.

Fusidic acid is generally not considered as a therapeutic alternative for the treatment of infections caused by Gram-negative rods. The agent was included in the study due to a susceptibility observed in vitro in some *A*. *actinomycetemcomitans* strains. The result of the study regarding fusidic acid susceptibility raised the question of whether this was a unique phenomenon for the species *A*. *actinomycetemcomitans*. Eleven strains of another *Aggregatibacter* species, *A. aphrophilus*, were included in the study to enable further evaluation. As shown in Figure 3, the fusidic acid MIC range for this bacterial species was 4–32 mg/L, similar to that found in the non-JP2 genotype *A*. *actinomycetemcomitans* (Figure 1b and Figure 3).

No obvious fusidic acid-resistant subpopulation was discovered by visual examination of the MIC distribution, except the single isolate with a MIC > 256 mg/L (Figure 1b), but the MIC range was broad and included values that were suggestive of resistance. Although the number of examined strains was low for an evaluation of differences in resistance, a tendency towards higher fusidic acid MICs was found in Swedish strains. When all non-JP2 strains isolated from Swedish and Ghanaian carriers were compared, 21 of 67 strains (31.3%) isolated in Sweden had fusidic MICs > 64 mg/L, but only 2 strains of 61 (3.3%) from Ghana, one of which was the highly resistant isolate (Table 3).

The fusidic acid MIC_50_ and MIC_90_ values for the 40 strains of *A. actinomycetemcomitans* of serotype a or c were 64 mg/L and 128 mg/L, respectively, in the 20 Swedish strains and 16 mg/L and 16 mg/L in 20 strains isolated from Ghanaian individuals. The impact of these strains on the MIC distributions is shown in Table 3. Interestingly there was a relationship between benzylpenicillin and fusidic MICs found in the strains of the study. The strains with low MICs for benzylpenicillin also had low fusidic acid MICs (Table 3). No correlation was seen between the MICs for benzylpenicillin and metronidazole (Appendix A). Strain J33, with a 640 bp deletion in the leukotoxin gene promoter region, had an MIC value for benzylpenicillin of 1 mg/L, and MIC of 4 mg/L for fusidic acid and metronidazole, which placed the strain well among the non-JP2 genotype strains (Table 2).

For metronidazole, the MIC values ranged from 0.5 to >256 mg/L (Figure 1c). The MIC distributions of the two genotypes merged and although the MIC values were generally lower for the JP2 genotype with MIC values that varied between 1 and 32 mg/L (Figure 1d), the MIC_50_ values of the two genotypes were within one two-fold dilution and thus not considered as significantly disparate (Table 2). The difference in MIC_90_ was caused by a subset of non-JP2 strains with high MIC values. Twenty-two non-JP2 genotype strains had metronidazole MIC values ≥ 64 mg/L (Figure 1c). These strains were found both among Swedish (15.2%) and Ghanaian strains (21.3%) and among all three examined serotypes (Table 4).

The MIC range for azithromycin contained seven two-fold dilution steps, but the MIC distribution did not reveal any obvious subpopulations (Figure 1d). The MIC range for tetracycline was 0.125–1 mg/L with a MIC_90_ of 1 mg/L (Table 1), and thus a narrow distribution with 90% of the strains within a MIC of 0.5 to 1 mg/L (data not shown). No bimodal distribution suggestive of the presence of a tetracycline-resistant subpopulation was detected by the visual examination of the MIC distribution.

## 3. Discussion

Few data are given in the literature on the antibiotic susceptibility of the JP2 genotype of *A. actinomycetemcomitans*. In the present study, the susceptibility of 32 JP2 genotype strains with various geographical origins, and 128 non-JP2 *A. actinomycetemcomitans* were tested against 11 antimicrobial agents. The susceptibility of the examined strains was analyzed in relation to MIC data, which revealed homogenous MIC distributions of strains with low or intermediate MIC values for most of the antimicrobials. High MIC_90_ values were found for metronidazole, benzylpenicillin and fusidic acid (Table 1). Interestingly, the MIC results for the latter two diverged between the JP2 genotype and the non-JP2 genotype with low, respectively high MICs for both antimicrobials as shown in Table 3. For benzylpenicillin and fusidic acid, a subpopulation containing the JP2 genotype strains, and six additional A. *actinomycetemcomitans* strains of serotype b had significantly lower MIC_50_ and MIC_90_ values than the rest of the strains (Table 2). No wild-type MIC distributions and thereby no ECOFFs are presented for *Aggregatibacter* on the EUCAST website (2022 v. 12.0) [31]. The MIC distribution for benzylpenicillin and the ECOFF calculated by the ECOFFinder with an estimation of 16 mg/L at 99.5% suggests that the majority of the non-JP2 genotype constitute a benzylpenicillin-resistant population (Figure 1a and Figure 2). On the other hand, in comparison with *Pasteurella multocida*, the only species in the family *Pasteurellaceae* with a defined clinical breakpoint for benzylpenicillin (≤0.5 mg/L/>0.5 mg/L) (EUCAST), all JP2 genotype strains of the study would be categorized as susceptible to benzylpenicillin. The benzylpenicillin PK/PD breakpoints are (≤0.25 mg/L/>2.0 mg/L) (EUCAST 2022 v. 12.0) [31], However, the AST of strains, and preferentially also a specific breakpoint for *Aggregatibacter*, is needed before benzylpenicillin can be recommended for the treatment of *A. actinomycetemcomitans* infections, and our data do not suggest peroral treatment of JP2 genotype carriers of *A. actinomycetemcomitans* with phenoxymethylpenicillin (penicillin V) or fusidic acid as a suitable therapy for local infections.

Differences in susceptibility against penicillins are usually due to acquired resistance mechanisms, such as the production of beta-lactamases or the presence or absence of variants of penicillin-binding proteins (PBPs). PBPs, which are enzymes aimed at the synthesis of the peptidoglycan layer of the bacterial cell wall, may have different binding affinities for penicillins [33,34]. Fusidic acid on the other hand exerts its activity by binding to intracellular structures, in staphylococci elongation factor G (EF-G), and thereby inhibits protein synthesis. Resistance occurs when genes encoding EF-G or protective mechanisms are altered [35]. In Gram-negative bacteria, permeability through the outer membrane and efflux mechanisms can also play a role in antibiotic susceptibility [36]. It is hard to envisage a common acquired resistance mechanism for the two agents, present in almost all the non-JP2 genotype strains. The generally higher fusidic acid MICs in the non-JP2 genotype of *A. actinomycetemcomitans* (Table 2), with an MIC range like that found in the *A. aphrophilus* strains, together with the higher MICs for benzylpenicillin, indicate that these susceptibility data reflect inherent properties, possibly in the cell wall structure in these groups of *Aggregatibacter.* Thus, the JP2 and the non-JP2 genotype may constitute two separate wild-type populations in relation to antimicrobial susceptibility. One MIC distribution for fusidic acid with a range between 2 and 64 mg/L is shown at the EUCAST website for *Haemophilus influenzae* [31], another member of the *Pasteurellaceae* family, also in agreement with what was found for the non-JP2 genotype of *A. actinomycetemcomitans* in the present study.

Six serotype b non-JP2 strains isolated in Sweden exhibited the same low MICs for benzylpenicillin and fusidic acid as the JP2 genotype (Figure 1a,b). We cannot exclude the possibility that some of these strains may originate from closely related subjects, but the strains were isolated from different years and both from individuals of African and Swedish descent. *A. actinomycetemcomitans* has by different techniques been divided in serotype correlated subpopulations [37,38], among which JP2 constitute a genotypically distinct group [11,39]. In the study by Haubek and co-workers, in which both multi-locus sequence typing (MLST) and restriction fragment length polymorphisms (RFLP) were used, a close relationship was seen between the JP2 genotype strains and some non-JP2 genotypes of serotype b reference strains [11]. We have not included a phylogenetic analysis on the examined collection of strains, but the six serotype b strains with low MICs for fusidic acid and benzylpenicillin are potentially related to such JP2-like serotype b bacteria.

The MIC distribution of metronidazole ranged from 0.5 to >256 mg/L and reflected great differences in the susceptibility among the examined strains (Figure 1c). As no clear-cut bimodal distribution was given, it is difficult to identify the wild-type population and a potential ECOFF value. A relatively high proportion of the non-JP2 strains, of all examined serotypes and both from Ghana and Sweden, had MIC values ≥ 64 mg/L, which suggests metronidazole resistance (Table 4). The MIC values for the 32 JP2 genotype strains were lower but still within a range that may include resistant strains (1–32 mg/L) (Table 2).

No ECOFFs are given for the metronidazole MIC distribution of anaerobic Gram-negative anaerobes on the EUCAST website, but the metronidazole MIC for 98% of the 3160 organisms in the *Bacteroides fragilis* group is at or below 4 mg/L. This is also the MIC value defined as the clinical breakpoint for metronidazole susceptibility in the anaerobic Gram-negative species *Bacteroides* and *Prevotella* [31].

*Aggregatibacter* is not phylogenetically related to anaerobic Gram-negative rods [40] and the effect of metronidazole on *Aggregatibacter* may be debated in view of the many studies that have shown high MIC values (Appendix A). However, at this breakpoint, 43% of the 160 examined *A. actinomycetemcomitans* strains of this study would be regarded as susceptible to metronidazole, but only 26% of the non-JP2 genotype strains. Of the 32 JP2 genotype strains, 7 (21.9%) would, with that assessment basis, have been deemed as resistant. Possible explanations of the heterogeneity in the MIC distribution for metronidazole and the wide range could be the presence of specific resistance mechanisms, differences in susceptibility between genotypes, or that the standardized AST influenced strains differently. Further evaluations are needed to make this clear. In the meantime, we suggest that the assumption that *A. actinomycetemcomitans* is susceptible to metronidazole is considered with caution.

Gentamicin, azithromycin, and tetracycline constituted a group with intermedium high MIC_90_ values of 1–2 mg/L (Table 1). No resistant subpopulation was detected by visual examination of the MIC distributions, exemplified by the distribution for azithromycin which had the widest MIC range (Figure 1d), but conclusions about whether *A. actinomycetemcomitans* should be regarded as susceptible, and the agents suitable for therapy, are difficult to draw in the absence of defined clinical breakpoints. The azithromycin MIC_90_ of the two genotypes differed significantly (*p* < 0.01) (Table 2) but the identical MIC_50_ values spoked against a more susceptible JP2 genotype, as did the MIC distribution (Figure 1c).

Previous studies of the antibiotic susceptibility of *A. actinomycetemcomitans* have mostly focused on potential therapies of local oral infections. We have in this study included antimicrobial agents intended for parenteral use in *A. actinomycetemcomitans* infections. Although the strains were collected from the oral cavity, the normal ecological niche for *A. actinomycetemcomitans*, we presume that they are representative also for invasive extra-oral infections, with the hypothesis that these bacteria also emanate from the oral flora, as has been suggested in studies of invasive *A. actinomycetemcomitans* non-oral infections [25,41].

In the present study, the MIC values for meropenem, cefotaxime, levofloxacin, and trimethoprim–sulfamethoxazole were all at or below 0.25 mg/L and amoxicillin, cefotaxime, meropenem, and levofloxacin had MIC values that would classify them as susceptible according to PK/PD breakpoints, indicating that the patient may be treatable with these antimicrobials, as well as with ampicillin. This assumption is supported by reports of cases of infective endocarditis caused by *A. actinomycetemcomitans* that were successfully treated with third-generation cephalosporins and with combinations of ampicillin plus gentamicin [41]. In patients with allergies to beta-lactam antibiotics, a quinolone may be considered for the treatment of non-oral *A. actinomycetemcomitans* infections. The low levofloxacin MIC range for the strains of this study (Table 1) corroborate earlier reports on low quinolone MICs in *A. actinomycetemcomitans* (Appendix A).

The effect of gentamicin is more difficult to envisage, but since no high MIC values for gentamicin were noted (Table 1), at least a potentiating effect of gentamicin on a given beta-lactam treatment may be possible to achieve. The JP2 genotype was described by Brogan and co-workers [15]. Strains of the JP2 genotype may have been included in examined collections of *A. actinomycetemcomitans* without having been identified as such, but to our knowledge the JP2 genotype has not been isolated from patients with endocarditis, bacteremia or other non-oral *A. actinomycetemcomitans* infections. Thus, according to the data of this study, the probability is high that these infections are caused by non-JP2 genotype strains with relatively high benzylpenicillin MIC values, and thus, questionable clinical effect of this agent.

The 160 examined strains in our study had an amoxicillin MIC range of 0.25–2 mg/L, indicating that they may constitute a population of wild-type bacteria, i.e., without acquired resistance mechanisms. The MIC range is within the range found by Jensen and co-workers when examining 257 strains, including a re-examination of eight strains previously reported to have high amoxicillin MIC values [28] (Appendix A). We did not test the combination of amoxicillin and clavulanic acid but, due to the lack of beta-lactamase production and the amoxicillin MIC values found, the strains of the study are interpreted as susceptible to amoxicillin/clavulanic acid according to the PK/PD breakpoints. However, as pointed out above, the same does not apply to benzylpenicillin and phenoxymethylpenicillin. Furthermore, AST would be required if penicillin treatment is considered in *A. actinomycetemcomitans*-associated periodontal infections, since not only the JP2 genotype of *A. actinomycetemcomitans* is associated with this disease [42]. The treatment of periodontitis primarily starts with mechanical debridement but as an adjunct antibiotic treatment may be considered, especially in more severe periodontitis (stage III–IV disease) [10,43,44]. When *A. actinomycetemcomitans* is assumed to participate in the infection, the recommended antibiotic treatment strategy has been a combination of amoxicillin and metronidazole [45]. Recent systemic reviews of reports of periodontitis treatment have shown a good clinical effect of this regiment, performed in addition to mechanical debridement [44,46]. However, studies evaluating the therapeutic effect of azithromycin, a potential adjuvant alternative, for example, in beta-lactam allergic patients, showed varying results [44]. In the present study, the MIC range for azithromycin was 0.064–4 mg/L, but higher values have been noted in other studies (Appendix A). Further studies are needed to evaluate if azithromycin resistance influences the clinical outcome of adjunctive treatment with azithromycin in patients with periodontitis.

The broad range of metronidazole MICs found in this study does not contradict the studies showing the good therapeutic effect of amoxicillin plus metronidazole for periodontitis, but rather support the identification of periodontitis as a polymicrobial oral infection [9]. None of the 160 *A. actinomycetemcomitans* strains of this study showed high amoxicillin MIC values or beta-lactamase production. The addition of a beta-lactamase inhibitor, for example, clavulanic acid, to amoxicillin could possibly be of value to overcome the influence of beta-lactamase production in concomitant oral bacteria susceptible to this combination [47].

## 4. Materials and Methods

### 4.1. Bacterial Strains

A total of 160 *A. actinomycetemcomitans* strains recovered from the same number of individuals were examined. Thirty-two strains were of the JP2 genotype, among them HK 1519, HK 1702, and HK 909, kind gifts from Dr. Mogens Kilian, Aarhus University, Denmark; LA 640 and LA 806 from Dr. Sirkka Asikainen, Umeå University, Sweden; HK 921 from Dr. Dorte Haubek, Aarhus University, Denmark; and P48 from Dr. Gunnar Dahlén, Gothenburg University, Sweden. The JP2 strain HK1651 (CCUG 56,173), isolated from a Ghanaian man living in Denmark, was from the Culture Collection, University of Gothenburg (CCUG), Sweden. Ten strains of the JP2 genotype were from individuals living in Ghana [48] and 13 were collected from individuals living in Sweden, four of whom were of African descent. Taken together, 18 of the 32 examined JP2 genotype strains originated from Africa (Appendix A). One strain, grouped with the non-JP2 genotype serotype b strains in this study, J33 (456A-13), was collected in Sweden from an individual born in Africa. It lacked the additional 110 bp in the leukotoxin promoter region, 108 upstream and 2 bp downstream of the 530 bp deletion found in the JP2 genotype [49]. Thus, by definition, strain J33 is not a JP2 genotype strain. In total, 67 of the 128 non-JP2 genotype strains were from individuals living in Sweden, and 61 were from Ghana. The non-JP2 strains were of serotype a, b, and c, the most prevalent serotypes reported from subgingival plaque samples [50] (Appendix A). The *Aggregatibacter aphrophilus* strains used in the study were HK372, 494A-13, 447A-13 and CCUG number 36,762, 49,493, 38,961, 43,422, 36,610, 32,956, 34,940, and 51,586.

### 4.2. Serotyping and JP2 Genotype Identification

*A. actinomycetemcomitans* strains collected from individuals living in Ghana and in Sweden were serotyped and leukotoxin-promoter-typed (JP2 genotype) at the clinical laboratory in Dental School, Umeå, Sweden, following the methods described by Höglund Åberg and co-workers [48]. Nine of the JP2 genotype strains were genotyped elsewhere.

### 4.3. Antibiotic Susceptibility Testing (AST)

Minimal inhibitory concentrations (MICs) of 11 antibiotics were tested with E-test gradient strips (bioMérieux Sweden AB, Askim, Sweden) according to the manufacturer’s recommendations. The tested antimicrobials were benzylpenicillin, amoxicillin (extended spectrum penicillin), cefotaxime (3^rd^-generation cephalosporin), and meropenem (carbapenem), which inhibit cell wall synthesis. Azithromycin (macrolide), gentamicin (aminoglycoside), and tetracycline inhibit protein synthesis. Levofloxacin (fluoro-quinolone) and metronidazole (imidazole) interfere with nucleic acid synthesis; and trimethoprim–sulfamethoxazole (trimethoprim–sulfonamide) inhibits tetrahydrofolic acid synthesis. Fusidic acid (fusidane), which inhibits protein synthesis, was included due to a sensitivity previously noticed by coincidence in some strains in vitro. To obtain confluent growth on the agar plates, bacterial suspensions were adjusted to McFarland 1.0 before plating on Fastidious Anaerobic Agar (FAA). The plates were incubated under aerobic conditions in the presence of 5% CO_2_ at 37 °C for 1–2 days, except for those with metronidazole strips that were incubated in an anaerobic glove box containing 85% N_2_/10% H_2_/5% CO_2_. These plates had been pre-reduced under anaerobic conditions before use. The resulting MICs were read at doubling dilution steps. Quality control was performed with *Bacteroides fragilis* (ATCC 25285) and *Haemophilus influenzae* ATCC 49766. The AST was performed in duplicate on separate occasions and repeated once if conflicting results were recorded.

### 4.4. Beta-Lactamase Production

Nitrocefin discs (BD BBL™ Cefinase™ β-Lactamase detection discs, Fisher Scientific, Göteborg, Sweden) were used for the detection of beta-lactamase.

*Staphylococcus aureus* ATCC 29213 was used as a positive control. Beta-lactamase-producing bacteria streaked on the disk give rise to a pink to red color.

### 4.5. Determination of Epidemiological Cut-Off Value (ECOFF)

The MIC distribution assessment for benzylpenicillin and the estimation of the ECOFF was performed with the freeware ECOFFinder, which estimates epidemiologic cut-off values based on nonlinear regression (https://clsi.org/meetings/microbiology/ecoffinder/accessed on 28 March 2021) [51].

### 4.6. Statistical Analysis

Data were assessed for statistical significance using Student’s *t*-test; a *p*-value less than 0.05 was used to establish significance.

## 5. Conclusions

In this study, we elucidated the antimicrobial susceptibility of strains of the JP2 and the non-JP2 genotype of *A. actinomycetemcomitans.* MIC values and MIC distributions were determined and thoroughly discussed. Taken together, it was concluded that the JP2 genotype strains, together with six additional serotype b strains, constitute a subpopulation more susceptible to benzylpenicillin and fusidic acid than the majority of non-JP2 genotype strains. This indicate that the JP2 genotype and the non-JP2 genotype may belong to different wild types in relation to antimicrobial susceptibility, but do not implicate per se that benzylpenicillin or fusidic acid is suitable for the treatment of infections associated with the JP2 genotype of *A. actinomycetemcomitans*. Low MIC values were noted for agents with potential for the parenteral treatment of invasive infections, such as cefotaxime, meropenem, levofloxacin and trimethoprim–sulfamethoxazole. None of the 160 tested *A. actinomycetemcomitans* had amoxicillin MIC values above 2 mg/L and the results of this study do not contradict the use of amoxicillin in combination with metronidazole, the established empiric antibiotic therapy of the polymicrobial infection severe periodontitis. Having the global increase in antibiotic resistance in mind, regular monitoring of the resistance status in oral bacteria may be of value, but due to the polymicrobial nature of severe periodontitis a routinely performed amoxicillin and metronidazole AST of only *A. actinomycetemcomitans* cannot be expected to reveal sufficient information to guide eventual therapy.

## Figures and Tables

**Figure 1 antibiotics-11-00317-f001:**
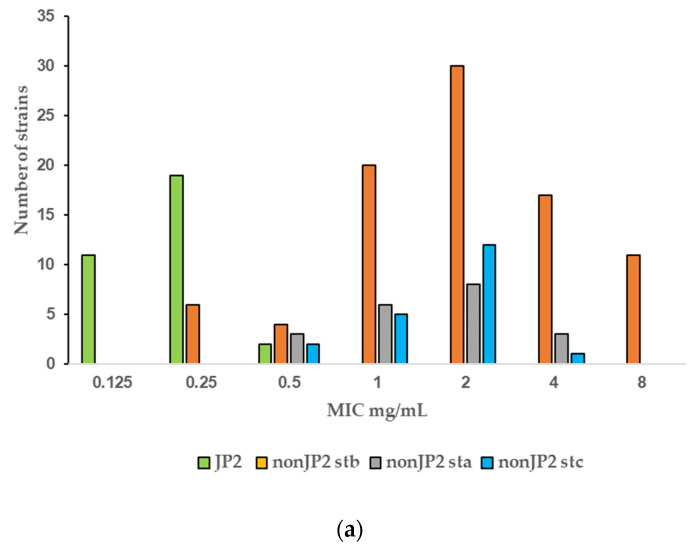
(**a**–**d**) MIC distributions of antimicrobials among JP2 and non-JP2 strains (mg/L). Strains: *n* = 160; JP2 genotype, *n* = 32; non-JP2 serotype b (stb), *n* = 88, non-JP2 serotype a (sta), *n* = 20; non-JP2 serotype c (stc), *n* = 20. (**a**) Benzylpenicillin. (**b**) Fusidic acid. (**c**) Metronidazole. (**d**) Azithromycin.

**Figure 2 antibiotics-11-00317-f002:**
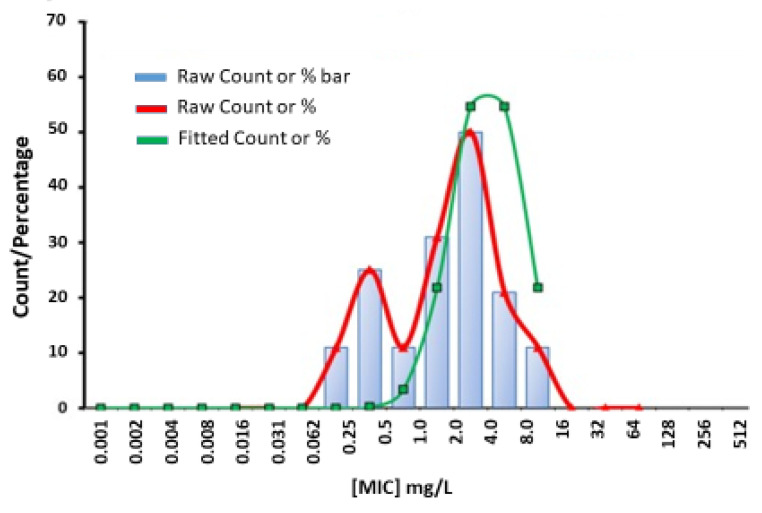
Estimation of wild-type MIC (mg/L) (0.064–16) distribution for benzylpenicillin against *A. actinomycetemcomitans* (*n* = 160) performed with ECOFFinder.

**Figure 3 antibiotics-11-00317-f003:**
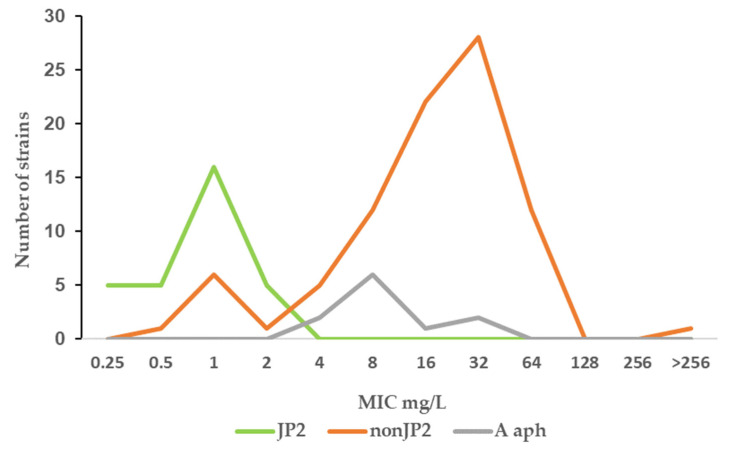
MIC distribution (mg/mL) of fusidic acid among *A. actinomycetemcomitans* JP2 strains (*n* = 32), non-JP2 strains (*n* = 88), and *A. aphrophilus* (*n* = 11).

**Table 1 antibiotics-11-00317-t001:** Antimicrobial susceptibility of 160 *Aggregatibacter actinomycetemcomitans*.

Antimicrobial	MIC Range (mg/L)	MIC_50_ (mg/L)	MIC_90_ (mg/L)
Benzylpenicillin	0.125–8	1	4
Amoxicillin	0.25–2	0.5	1
Cefotaxime	<0.016–0.25	0.064	0.064
Meropenem	0.064–0.25	0.125	0.125
Azithromycin	0.064–4	0.5	1
Fusidic acid	0.25–>256	16	64
Gentamicin	0.25–4	1	2
Levofloxacin	<0.002–0.16	0.004	0.008
Metronidazole	0.5–>256	4	128
Tetracycline	0.125–1	0.5	1
Trimethoprim–Sulfamethoxazole	<0.002–0.064	0.008	0.032

**Table 2 antibiotics-11-00317-t002:** MICs of metronidazole, benzylpenicillin, azithromycin, and fusidic acid in *A. actinomycetemcomitans* strains of the JP2 (*n* = 32) and the non-JP2 genotype (*n* = 128).

Antimicrobial	MIC-Range (mg/L)	MIC_50_ (mg/L)	MIC_90_ (mg/L)	*p*-Value *
Metronidazole				
JP2	1–32	4	16	<0.001
non-JP2	0.5–>256	8	128	
Benzylpenicillin				
JP2	0.125–0.5	0.25	0.25	<0.001
non-JP2	0.25–8	2	4	
Azithromycin				
JP2	0.064–1	0.5	0.5	<0.01
non-JP2	0.064–4	0.5	2	
Fusidic acid				
JP2	0.25–2	1	2	<0.001
non-JP2	0.5–>256	32	64	

* Calculated with *t*-test for difference between MIC_90_ values.

**Table 3 antibiotics-11-00317-t003:** The relationship between MIC values for benzylpenicillin (BP) and fusidic acid in 160 *Aggregatibacter actinomycetemcomitans* strains.

BP MICmg/L	Fusidic Acid, MIC mg/L
0.25	0.5	1	2	4	8	16	32	64	128	≥256
8							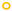	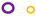	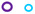		
4						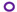	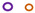	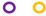	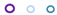	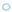	
2					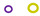	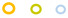	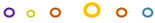	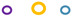	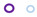	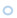	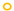
1				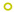	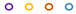	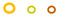	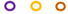	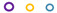	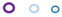		
0.5			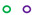	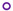	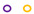	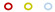	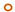		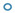		
0.25	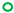	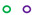	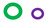	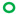							
0.125	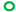	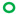	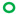								

The origin of strains is represented by colours and the numbers by the size of the circle. 
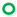
 JP2 genotype strains (*n* = 32); 
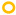
 serotype b strains (Ghana, *n* = 41); 
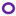
 serotype b strains (Sweden no. 47); 
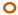
 serotype a strains (Ghana, *n* = 10); 
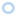
 serotype a strains (Sweden, *n* = 10); 
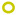
 serotype c strains (Ghana, no. 10); 
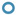
 serotype c strains (Sweden, *n* = 10). The size of the circle in relation to number of strains: 1: 

; 2: 
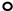
; 3: 
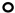
; 4: 
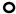
; 5: 
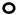
; 6: 
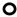
; 7: 
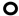
; 8: 
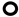
; 9: 
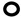
; 10: 
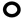
; 11: 
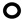
; 12: 
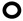
.

**Table 4 antibiotics-11-00317-t004:** Metronidazole susceptibility in non-JP2 strains isolated from carriers from Sweden and Ghana.

Origin and Serotype	Number	MIC ≥ 64 mg/L (Number)	Highly Resistant Strains (%)
Sweden, b	47	10	15.2
Sweden, a + c	20	0	
Ghana, b	41	6	21.3
Ghana, a + c	20	7	

## Data Availability

The raw data supporting the conclusions of this paper will be made available by the authors without undue reservation.

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
