# Peer review of "Discrepancies in Antimicrobial Susceptibility between the JP2 and the Non-JP2 Genotype of Aggregatibacter actinomycetemcomitans"

_antibiotics, 2022, doi:10.3390/antibiotics11030317_

Round 1
Reviewer 1 Report
Major issues:
Section 4.3: How many times was each strain tested with each antibacterial drug/strip? It is insufficient to test each strain only once with each drug, and nothing in the manuscript suggests that this was not the case.
Line 193: Table 3 is a table, not a scatter plot. Additionally, these data being reported in table form makes it very hard to interpret. These data should be presented in a bubble chart. This would allow visualization of the relationships and the number of strains at each point. Also, it should be stated in the caption that BP stands for benzylpenicillin.
Although the conclusions are generally scientifically sound and of interest, the manuscript is somewhat hard to read and follow because there is often little or no explanation of the rationale when moving from one figure to the next.
Minor/specific issues:
Line 75: AST has not been used previously, so it should be defined.
Line 71-97: Given the generally broad audience of this journal, it would be beneficial to briefly explain/define breakpoints and differences between divergent breakpoints, clinical breakpoints, PK/PD breakpoints, etc.
Line 84: EUCAST has not been previously used, so it should be defined.
Lines 116-124: This paragraph is somewhat hard to follow, potentially due to its organization. Why were the four drugs in line 120 chosen for additional investigation? If it was due to the statement in lines 121-124, then this statement should start the paragraph to set up and explain what was done next and why it was done next.
Line 129: The wording suggests that the data shown in Figure 1b for fusidic acid was broken down by serotype as was done in Panel A for benzylpenicillin; however, this was not actually the case. Also, why is Panel A the only one broken down into serotypes?
Lines 178: Why was A. aphrophilus chosen/introduced here?
Figure 3: Why is Figure 3 a line graph rather than a bar graph like Figure 1? It would be much more clear if it were a bar graph like Figure 1, and then the data that was omitted “for clarity” (lines 177-179) could easily be included.
Line 187: It is not clear what drug this data is for.
Line 189: The previous sentence implies that data for 20 Swedish and 20 Ghanaian strains are included in Table 3; however, the table itself indicates that there were 66 Swedish and 41 Ghanaian strains examined.
Line 195: Please include this data in the supplement, rather than omitting it.
Line 216: Since Figure 1c is talked about after Figure 1d, these two panels should be reversed in the figure.
Line 276: MLST and RFLP have not been used previously, so they should be defined.
Lines 390-394: Why was J33 classified as non-JP2 when it has the same region deleted (plus additional bases on either side) as the JP2 genotype?
Author Response
Comments to Reviewer 1.
Major issues 1-3:
1.Section 4:3. How many times was each strain tested with each antibacterial drug/strip?
The following sentence is now included in the section for Material and methods, on line 471: The AST were performed in duplicate on separate occasions and repeated once if conflicting results were recorded. (459)
- Line 193. Table 3 is a table not a scatter plot. Additionally, these data being reported in table form makes it very hard to interpret. These data should be presented in a bubble chart. This would allow visualization of the relationships and the number of strains at each point. Also, it should be stated in the caption that BP stands for benzylpenicillin.
We have given the data shown in table 3 another presentation that we hope is more illustrative. However, the presentation is not in the form of bubble charts since they, in our hands, not fulfilled our purpose in relation to the presentation of the results. (211)
- Although the conclusions are generally scientifically sound and of interest, the manuscript is somewhat hard to read and follow because there is often little or no explanation of the rationale when moving from one figure to the next.
With the aim to improve this we have made changes in the text under the results section starting with lines 118, 189 and 202 in the primary submitted paper.
Line 118: Included sentences: Four antimicrobials; metronidazole, benzylpenicillin, fusidic acid, and azithromycin, showed wide MIC-ranges and/or high MIC90 -values (Table 1). In Figure 1 we present MIC distributions for these antimicrobials (Fig 1). (128)
Line 176. Included sentences: Fusidic acid is generally not considered as therapeutic alternative for the treatment of infections caused by Gram-negative rods. The agent was included in the study due to an in vitro observed susceptibility in some A. actinomycetemcomitans strains. The result of the study regarding fusidic acid susceptibility raised the question whether this was a unique phenomenon for the species A. actinomycetemcomitans Eleven strains of another Aggregatibacter species, A. aphrophilus, were included in the study to enable further evaluation. As shown in figure 3 the fusidic acid MIC-range for this bacterial species was 4 – 32 mg/L, similar to what was found in non-JP2 genotype A. actinomycetemcomitans (Figure 3) (Figure 1b). (183)
Line 187: This paragraph has been given another structure (198 – ).
No obvious fusidic acid resistant subpopulation was discovered by visual examination of the MIC distribution, except the single isolate with a MIC > 256 mg/L (Figure 1b), but the MIC range was broad and included values that were suggestive of resistance. Although the number of examined strains were low for an evaluation of differences in resistance a tendency for higher fusidic acid MICs was found in Swedish strains. When all non-JP2 strains isolated from Swedish and Ghanaian carriers were compared 21 of 66 strains. When all nonJP2 strains isolated from Swedish and Ghanaian carriers were compared 21 of 66 strains (31.8%) isolated in Sweden had fusidic MICs > 64 mg/L, but only two strains of 61 (3,3%) from Ghana, one of which was the highly resistant isolate (Table 3).
The fusidic acid MIC50 and MIC90 values for the 40 strains of A. actinomycetemcomitans of serotype a or c were 64 mg/L and 128 mg/L, respectively, in the 20 Swedish strains and 16 mg/L, and 16 mg/L in 20 strains isolated from Ghanaian individuals. The impact of these strains on the MIC distributions is shown in Table 3.
Interestingly there was a relation between benzylpenicillin and fusidic MICs found in the strains of the study. The strains with low MICs for benzylpenicillin also had low fusidic acid MICs (Table 3). No correlation was seen between the MICs for benzylpenicillin and metronidazole, another agent with a wide MIC-range (Table S3). Strain J33, with a 640 bp deletion in the leukotoxin gene promoter region, had a MIC value for benzylpenicillin of 1 mg/L and MIC 4 mg/L for fusidic acid and metronidazole respectively, which placed the strain well in among the non-JP2 genotype strains (Table 2).
The relations between benzylpenicillin and fusidic MICs in the strains of this study are shown in (Table 3). A correlation found between benzylpenicillin and fusidic MICs in non-JP2 strains isolated in Sweden and Ghana are shown in the scatter plot in Table 3, together with the data of all JP2 strains (Table 3). The strains with low MICs for benzylpenicillin also had low fusidic acid MICs. No correlation was seen between the MICs for benzylpenicillin and metronidazole (data not shown). Strain J33, with a 640 bp deletion in the leukotoxin gene promoter region, had a MIC value for benzylpenicillin of 1 mg/L and MIC 4 mg/L for fusidic acid and metronidazole respectively, which placed the strain well in among the non-JP2 genotype strains (Table 2).
Minor issues/
Line 75. AST has not been used previously, so it should be defined.
It has been defined as antimicrobial susceptibility testing (AST) (76)
Line 71-95. Given the generally broad audience of this journal, it would be beneficial to briefly explain/define breakpoints and differences between divergent breakpoints, clinical breakpoints, PK/PD breakpoints etc. It would be beneficial to briefly explain/define breakpoints and differences between divergent breakpoints, clinical breakpoints, PK/PD breakpoints etc.
The following sentences has been included. We hope this change in conjunction with the next paragraph will meet the request.
Antimicrobial clinical breakpoints are determined by breakpoint committees and constitute concentrations of antibiotics that for an identified bacterial infection will guide in the choice of antibiotic treatments [29,30]. Many factors are taken into account in the process of setting breakpoints, for example pharmacokinetic/pharmacodynamic (PK/PD) relationships for the antimicrobial agents, type and location of the infection, dosing regiments, toxicology, resistance mechanisms and wild-type MIC distributions. The definitions of both clinical breakpoints and non-species related PK/PD breakpoints are ongoing processes in the European Committee on Antimicrobial Susceptibility Testing (EUCAST), and changes of the breakpoints are published yearly on the website [31].
Line 84. EUCAST has not been used previously, so it should be defined
It has now been defined as European Committee on Antimicrobial Susceptibility Testing (EUCAST) (86)
Lines 116-124. This paragraph is somewhat hard to follow, potentially due to its organization. Why were the four drugs in line 129 chosen for additional investigation? If it was due to the statement in lines 122-124, then this statement should start the paragraph to set up and explain what was done next and why it was done next.
The paragraph now reads: Four antimicrobials; metronidazole, benzylpenicillin, fusidic acid, and azithromycin, showed wide MIC-ranges and/or high MIC90 -values (Table 1).
In Figure 1 we present MIC distributions for these antimicrobials (Fig. 1) (128).
Line 142 the figure reference should be (Figure 1a, and 1b) (138)
Line 129. The wording suggests that the data shown in Figure 1b for fusidic acid was broken down by serotype as was done in Panel A for benzylpenicillin; however, this was not actually the case. Also, why is Panel A the only one broken down into serotypes?
We have included serotypes in all the panels of figure 1. (153-170)
Line 178. Why was A. aphrophilus chosen/introduced here?
Please see the response for line 189 in the revised version.
Figure 3: Why is Figure 3 a line graph rather than a bar graph like Figure 1? It would be much more clear if it was a bar graph like Figure 1, and then the data that were omitted for “clarity” (lines 177-179) could easily be included.
The purpose of figure 3 is primarily to show the data for A. aphrophilus in relation to those for the JP2 and non-JP2 genotype strains of A. actinomycetemcomitans. We think the line graph illustrate that well (192)
The data for all 160 strains of A. actinomycetemcomitans are shown in figure 1, now with all studied serotypes include. (153-170)
Line 187: It´s not clear what drug this data is for.
It should be Fusidic acid. Please see the rephrasing done in that paragraph as documented under major issue 3.
Line 189. The previous sentence implies that data for 20 Swedish and 20 Ghanaian strains are included in Table 3; however, the table itself indicates that there were 66 Swedish and 41 Ghanaian strains examined.
Table 3 is now presented in a different way, including all 160 strains (211-222) The bacteria in the study were from Sweden and Ghana and of serotype a, b, and c as documented in Material and methods and Table S1. The ambiguity will hopefully disappear with the change of table 3, see above under major issue 2.
Line 195.Please include these data in the supplement rather than omitting it.
The data are now presented in the supplement Table S3.
Line 216. Since Figure 1c is talked about after Figure 1 d, these two panels should be reversed in the figure.
The two have been reversed. (251)
Line 276. MLST and RFLP have not been used previously, so they should be defined.
The two have been defined: multi-locus sequence typing (MLST), Restriction fragment length polymorphisms (RFLP). (311)
Lines 390-394. Why was J33 classified as non-JP2 when it has the same region deleted (plus additional bases on either side) as the JP2 genotype?
The definition of the JP2 genotype is that it harbors a 530 nucleotide base pair deletion located in the leukotoxin promoter region. The occurrence of another deletion in the same region of the genome could be explained by a hot spot for recombination or a common insertion place for mobile genetic elements, as insertion sequences, in that part. Any of these phenomena can possibly occur in Aggregatibacter actinomycetemcomitans with mutually different genetic backgrounds.
Reviewer 2 Report
The paper is interesting from both microbiological and clinical point of view as its results confirmed that A. actinomycetemcomitans may be divided into two separate wild-types, with a suggested intrinsic reduced susceptibility for benzylpenicillin in the majority of non-JP2 genotype strains and this finding could have important implications for proper treatment of A. actinomycetemcomitans infections. The applied methods are reliable, the conclusions are supported by the obtained results and the references are relatively actual, properly selected and used. In my opinion the paper could be published in Antiobiotics after including a description of the methods of statistical analysis of the obtained results at the end of the chapter Materials and Methods.
Author Response
The methods of statistical analysis have been included at the end of Material and methods.
Line
4.6 Statistical analysis. Data were assessed for statistical significance using Student´s t-test; a p-value less than 0.05 was used to establish significance (473).
Round 2
Reviewer 1 Report
Prior concerns were sufficiently addressed. Four editing issues are noted below that should be corrected prior to publication.
Table 3: In the pdf version of the revised manuscript that I downloaded, the circles in the table do not appear to have any variation in size and all appear to match the circle size 1 given in the scale below the table. This is an improvement over the previous representation of this data, but the technical issue with how the table uploaded/circle sizes in the table should be addressed and corrected.
Lines 202 and 280: update Figure 1 reference to reflect the re-ordering of the panels - this should now be Figure 1c, rather than Figure 1d.
Line 214: update Figure 1 reference to reflect the re-ordering of the panels - this should now be Figure 1d, rather than Figure 1c.
In the author's response to the initial comments, they stated that they had added the following sentence to address the number of times that each strain was tested; however, the sentence could not be located in the version of the manuscript that I received. "The following sentence is now included in the section for Material and methods, on line 471: The AST were performed in duplicate on separate occasions and repeated once if conflicting results were recorded. (459)"